# Resilience, ingenuity, and identity: A multi-level analysis of the Filipino community health worker experience in rural and remote municipalities in the Philippines

Regine Ynez H. De Mesa[1,2]*, Zoé Mistrale Hendrickson[1], Carol Stephanie C. Tan-Lim[2,3], Anton Elepaño[4], Noleen Marie C. Fabian[2,5], Johanna Faye E. Lopez[2], Carl A. Latkin[1], Leonila F. Dans[2,6], Mia P. Rey[2,7], Antonio Miguel L. Dans[2,8]

1 Department of Health, Behavior, and Society, Johns Hopkins Bloomberg School of Public Health: Johns Hopkins University Bloomberg School of Public Health, Baltimore, Maryland, United States of America, 2 Philippine Primary Care Studies, University of the Philippines Diliman, Quezon City, Philippines, 3 Department of Clinical Epidemiology, College of Medicine, University of the Philippines Manila, Manila, Philippines, 4 Nuffield Department of Primary Care Health Sciences, University of Oxford, Oxford, United Kingdom, 5 College of Nursing, University of the East Ramon Magsaysay Medical Center, Quezon City, Philippines, 6 Department of Pediatrics, College of Medicine, University of the Philippines Manila, Manila, Philippines, 7 Department of Accounting and Finance, Virata School of Business, University of the Philippines Diliman, Quezon City, Philippines, 8 Health Equity and Research Foundation, St. Luke's Medical Center – Global City, Taguig City, Philippines

* rhdemesa.ppcs@gmail.com

## Abstract

Community Health Workers (CHWs) have an extensive involvement in augmenting service capacities in primary care settings. This study sheds light on the unique experiences of CHWs as they navigate barriers and enablers in the Philippine healthcare setting through their journey for professionalization. This study aims to: (1) Describe the roles assumed by CHWs in rural and remote municipalities in the Philippines; and (2) Identify the multi-level barriers and enablers CHWs perceive to influence their performance of these roles. From June to July 2023, the Philippine Primary Care Studies parent program piloted a study on a clinical decision support tool for CHWs, involving 34 CHWs across six focus group discussions. The interviews also touched upon the roles of CHWs and the factors influencing their performance within their local health settings in-depth. A mixed inductive/deductive approach was used to investigate this subset of the FGD data. CHWs assume diverse roles that often surpass health service provision. While their roles were crucial, CHWs described being positioned against a volatile political landscape fraught with material insecurity. They utilized their individual and interpersonal capacities to overcome situational limitations and were augmented with organizational level interventions like improved network connectivity, training, or expanded access to clinical decision-support tools. Amidst resource scarcity, CHWs demonstrated remarkable resilience through their own ingenuity and by maximizing support from their social networks. While their commitment

**Data availability statement:** All relevant data are within the manuscript and its Supporting information files.

**Funding:** The study was funded by the National Academy for Science and Technology (NAST). No grant number was provided by the funder. Funding was not awarded to any individual author. This grant was directed to the Philippine Primary Care Studies. The funder had no role in study design, data collection and analysis, decision to publish, or preparation of the manuscript. JFL and NCF received a salary from NAST as compensation in their efforts to collect data for the parent study. RDM and AE received compensation from the same funder in their role as lead and assistant study analysts respectively.

**Competing interests:** The authors have declared that no competing interests exist.

is an asset to the health workforce, support from national policymakers and local governments units are crucial to ensure CHWs remain protected against systemic exploitation. Ensuring accountability and stronger implementation of pre-existing laws to ultimately recognize the role of CHWs are an essential way to support CHWs and improve community health.

## 1. Introduction

### 1.1. Background of the study

Community health workers (CHWs) are pivotal to augmenting primary care service capacities in low and middle-income countries [1]. In settings like the Philippines, integrating CHWs into healthcare provider networks has been a strategic response to address service demands amid increasing workforce shortages [2]. CHWs have been a salient feature of the healthcare system in the Philippines for decades, with each CHW responsible for the welfare of residents in their assigned communities. As of 2020, an estimated 241,282 CHWs were reported to be in active service [3]. Locally referred to as Barangay Health Workers, named after the *barangay*—the smallest administrative unit in the Philippines, CHWs have become embedded in the delivery of primary care services since the program's inception.

The recognition of lay health workers in national healthcare frameworks gained momentum following the Alma Ata Declaration of 1979. In response, the Philippines established primary care as a national priority and subsequently launched its CHW program in 1981 [4,5]. Existing literature suggests that CHWs assume a minor role in barangay health centers as their ability to participate in direct service provision is contingent on whether midwives or nurses are present to supervise them [6,7]. Nevertheless, the role of CHWs extends beyond healthcare facilities. Their duties include providing culturally appropriate medical advice, facilitating referrals, and mediating between patients and providers [8]. Although increasingly recognized for these bridging functions, the full range of CHW responsibilities is still largely dependent on local leadership [1,4,9]. Amidst fragmentation in health governance and policy, a comprehensive description that captures the full scope of CHW work becomes all the more elusive.

Signaled by the passage of Republic Act No. 7160 or the Local Government Code of 1991, national entities delegated a considerable portion of healthcare decision-making authority to local government units (LGUs) like the municipality or the barangay [10]. Under this set-up, LGUs take charge of operating the primary, secondary and tertiary levels of care. The primary level focuses on patients in the community setting while the latter levels take place in hospitals [4]. CHWs operate on the primary level. They collaborate with other members of the healthcare provider network in health promotion and disease prevention efforts. Despite this standardized set-up, systems devolution has contributed to the variability and expanded breadth of CHWs' responsibilities [4]. Decisions on the procurement of medical equipment, health budget allocation, and the onboarding of health personnel were principally

determined by these localized units. Within a decentralized framework, the recruitment of CHWs, the honorarium set aside for them, and the specific duties assigned to CHWs are all under the purview of their respective LGUs [6]. While devolution aimed to bolster system responsiveness, actualizing this goal still hinges on the commitment of the incumbent leadership at the local level. This precariously positions CHWs against a potentially dynamic political landscape, as their employment and the resources they have access to are predicated on the priorities of the LGU [1].

CHWs have been operational in barangay health centers longer than their counterparts in other countries [6,11]. However, the only legislative measure safeguarding their rights and welfare was enacted through the BHWs' Benefits and Incentives Act of 1995. This act aimed to protect CHWs' right to self-organize and mandated LGUs to provide a form of subsistence allowance [12]. In late 2022, House Bill No. 6557 or the Magna Carta for Barangay Health Workers subsequently received unanimous approval in its conclusive third reading [13]. The passage of the Magna Carta marked a significant step towards ensuring CHWs receive workplace protections, which include health benefits, retirement cash incentives, and hazard pay. With the passage of the Philippines' Universal Health-care Law of 2019, LGUs are slated to receive capitated funds from the Philippine Health Insurance Corporation to support primary care services [14]. If these funds are implemented with system-strengthening interventions as intended, augmented organizational support can alleviate the burden on CHWs to address resource scarcity on an individual level. On ground realities, however, are far from these goals. Full implementation of these laws have stalled due to varying health priorities among LGUs, the lack of monitoring systems to ensure adherence [4], and inadequate awareness among CHWs themselves [15]. Moreover, protections specifying the permissible scope of work for CHWs are lacking, the grounds for lawful termination are ambiguous, and compensatory practices are largely discretionary [1].

The inadequate attention to CHWs in legislation is symptomatic of the institutional undervaluation of their role in improving community health outcomes. A nationally representative survey reported that over 50% of health visits from the country's poorest wealth quintile were made to community health centers [16]. In underserved areas, these facilities were primarily staffed by CHWs who lacked medical supplies [6] and training to support increasing service demands. CHWs are often the first and sometimes the only point of healthcare contact for many of these community frontlines. Despite their involvement in augmenting service capacities, existing research on the roles they assume and the barriers and enablers that influence their role performance remains limited. Therefore, examining the multi-level factors that influence role performance is crucial in ensuring policies respond to the experiences of CHWs in the field.

### 1.2. Study objectives

The present study aims to achieve two objectives: (1) To describe the roles assumed by CHWs in a rural and remote municipality in the Philippines; and (2) To identify the multi-level barriers and enablers CHWs perceive as influencing the performance of these assumed roles.

## 2. Methodology

### 2.1. Research gap

CHWs have been the backbone of the Philippine healthcare system for over four decades. Despite the passage of policies aimed at improving workplace protections and financial support, CHWs continue to encounter persistent challenges in carrying out their roles. These challenges are further shaped by the decentralized structure of the healthcare system, making them highly context-specific. This presents a critical research gap. Exploring the enablers and barriers to CHW role performance through their lived experiences offers valuable, grounded insights into the realities of service delivery in devolved settings. Such understanding is essential for informing policies that are responsive to realities at the grassroots level. Focus group discussions (FGDs) were conducted to capture nuanced perspectives. This method enables rich, in-depth exploration of shared experiences within their local contexts.

## 2.2. Conceptual framework

Using the socioecological model adopted from Bronfenbrenner's ecological systems theory of development, this study examines the factors that enable or inhibit CHW role performance across various spheres of influence [15,16]. The model asserts that individual, interpersonal, organizational, and policy-level factors influence human experience or, in this case, role performance [17,18]. Our framework adopts Gjerde & Ladegard's conceptualizations [19], defining 'roles' as the spectrum of behaviors and responsibilities incumbent upon individuals within an organization. To examine factors influencing role performance, we have integrated insights from Mhlongo et al., [20] who characterize barriers as elements that adversely impact CHWs' ability to perform their roles, and enablers as factors that facilitate this performance. Fig 1 presents the framework and working definitions used in this study [19,20].

   Despite the paucity of studies exploring CHW experiences in the Philippines, existing research has leaned on analytical frameworks like Campbell and Cornish's triad of social contexts [1,4] or Bronfenbrenner's multi-level approach [6]. The present analysis gravitated towards the latter to avoid predetermining more specific thematic categories and to clarify the level at which interventions should be directed. Community and policy-level factors were collapsed into a single umbrella category for parsimony.

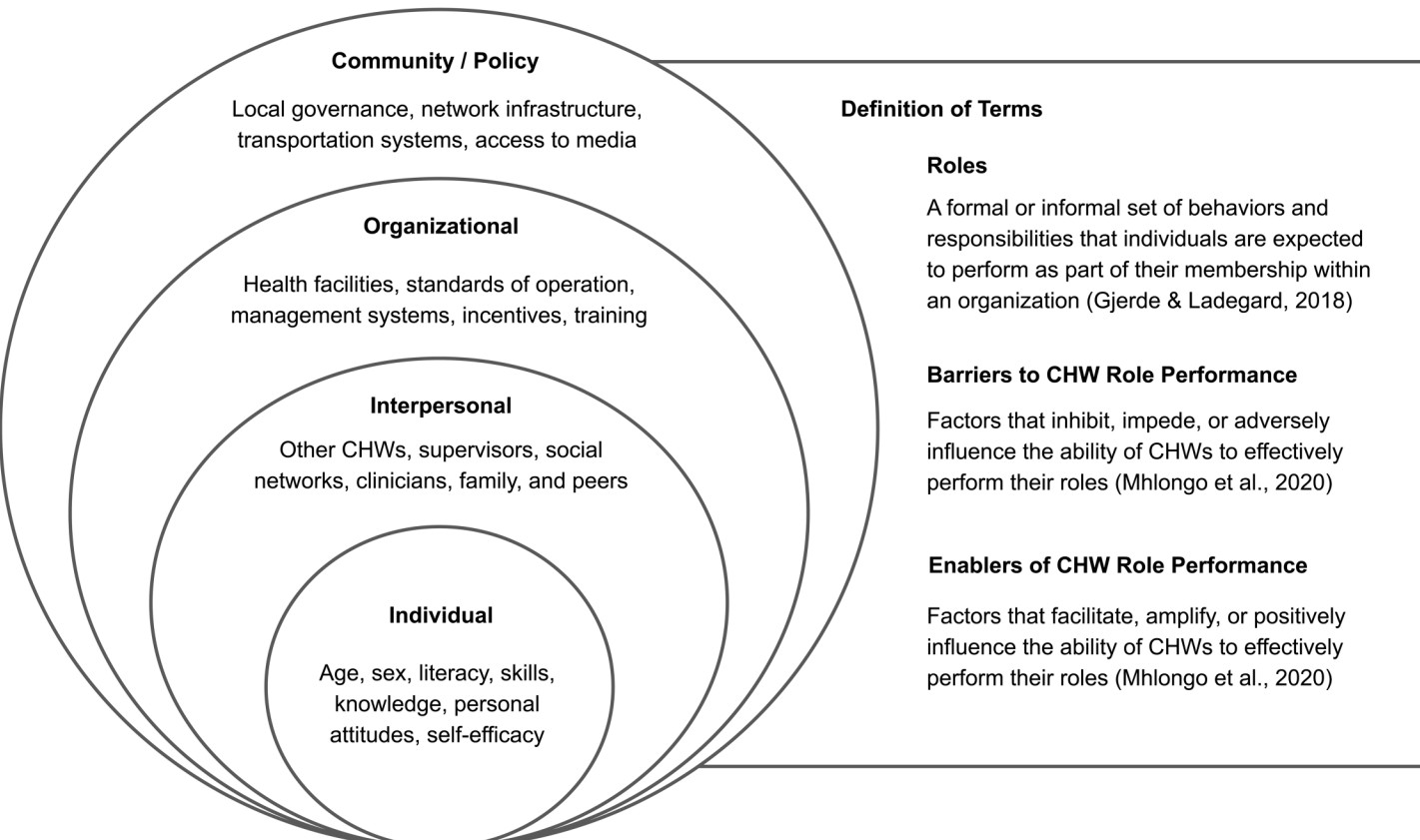

**Fig 1. Adaptation of Bronfenbrenner's socioecological model and definitions of roles, barriers, and enablers.**

## 2.3. Study setting

This study presents a secondary analysis of focus group data obtained by the Philippine Primary Care Studies (PPCS) program. PPCS is a publicly-funded program aimed at strengthening primary care systems through universal health coverage and building workforce capacities [21]. The parent study, from which our data originated, focused on exploring the use of a clinical decision support tool within a cadre of CHWs. The parent study for this analysis was conducted in two rural and remote municipalities in the Philippine provinces located in Central and Southern Luzon, respectively. Given the politically sensitive narratives presented in this study, the researchers opted to anonymize the specific municipalities and barangay units from which participants were recruited.

According to the 2020 National Census, the rural site in Central Luzon supports an estimated total population of 853,373 people across 11 municipalities and one city [20,21]. In comparison, the remote site in Southern Luzon is, over 360 miles from the Philippines' National Capital Region, a Pacific-facing peninsula home to an estimated population of 828,655 people across 14 municipalities and one city [22]. The province of our remote site features a mountainous landscape and is particularly prone to typhoons, owing to its position along the Pacific typhoon belt [21,23,24]. CHWs in both provinces have been the backbone for supporting essential primary care services and public health initiatives.

## 2.4. Sampling and data collection

Recruitment for the parent study began on June 11, 2023 and concluded on July 7, 2023. Six focus group discussions (N = 6) with approximately five to eight participants were conducted at each site. CHWs were purposively recruited to reflect various lengths of service in their field. Three tenure categories were used to diversify participant recruitment, broadly defined as s*hort tenure* for those with less than three years of experience, *medium tenure* for individuals who had been in their roles between three to six years, and *long tenure* for those with over six years of experience as a CHW. These tenure cutoffs were aligned with the general quartile distribution of the years in service of CHWs in rural and remote areas, which were last measured by the PPCS program in 2022.

Inclusion criteria for recruitment encompassed CHWs currently employed by each municipality, individuals 18 years and above, and those with prior exposure to the clinical decision support tool piloted by the parent study. The inclusion of CHWs with prior exposure to UpToDate was crucial for the usability study undertaken by the parent study. This approach ensured that participants were already familiar with or had experience using the application, enabling them to discuss their perceptions of the tool. CHWs who had retired before data collection commenced were excluded from recruitment. A combined total of 34 individuals were recruited and participated across both sites.

A multi-lingual team of researchers conducted all focus group discussions. Focus groups were facilitated in English and Filipino in the rural site and English and Bikolano, a local language spoken in the remote site. Participants in these settings used each of these language pairs interchangeably as part of their vernacular. Focus group sessions lasted between 60–90 minutes and were audio-recorded by researchers of the parent study. While the parent study did not provide any financial incentives, participants were reimbursed for transportation costs and received groceries as a token of appreciation. Both verbal and written informed consent were obtained from participants before each focus group session.

## 2.5. Focus group discussion guide and compatibility

The parent study adopted a semi-structured focus group discussion guide to assess the compatibility of UpToDate in supporting CHW workflows [25]. Pilot testing for the adapted study instrument was conducted by the parent study in the same two sites and an additional urban site in April 2021. In addition to investigating user perceptions of UpToDate, this adapted facilitator's guide included questions that delved into CHWs' roles and capacity-building needs. FGDs for the parent program touched upon the experiences of CHWs and the factors influencing their performance within their local health units in-depth. For example, the CHWs were asked about the scope of their daily responsibilities, the command

hierarchies within their practice, and the various types of resources or support available to them. The facilitator's guide also had prompts exploring perceived barriers and enablers of effective role performance among CHWs (see S1 Appendix). A mixed inductive/deductive approach was used to investigate this subset of the FGD data for secondary analysis. Since participants recruited for the present study included English, Filipino, and Bikolano speakers, the interview guide was translated by the facilitators into the language/s spoken by the participants.

## 2.6. Data management

Three research assistants from the parent study transcribed each interview verbatim before translating it into English. All personally identifiable information, such as names, specific barangay assignments, or identifiable participant attributes were appropriately redacted during transcription. Subsequently, each participant was assigned an unlinkable case ID before initiating secondary analysis. Anonymized data was accessed by analysts for research purposes on September 1, 2023. The secondary analysis described in this paper was conducted by a social scientist and a medical doctor with prior exposure working in each of the settings described in Section 2.2. Neither analyst was directly involved in collecting or transcribing focus group data under the parent study. Furthermore, all names presented in this study are pseudonyms and bear no relation or resemblance to the actual names of the participants.

## 2.7. Data analysis

All verbatim transcripts from the six focus groups were uploaded into NVivo 14.0 for analysis. The present study used a mixed deductive and inductive approach to develop the codebook and organize the data into themes. Deductive codes involved an *a priori* interest in the roles assumed by CHWs and the factors that either impeded or facilitated their role performance. In this study, we employed open and axial coding for our qualitative data analysis. Open coding involved segmenting narrative data into thematic components, while axial coding focused on identifying relationships between these components and organizing them into broader categories.

Inductive codes were elicited by randomly drawing two transcripts and having two researchers independently identify emergent codes (open coding). To do this, the secondary data analysis team reviewed the untranslated transcripts and their corresponding English translations while coding.

After open coding, both analysts convened to develop a preliminary codebook wherein codes were either collapsed or expanded. CHW roles, as organized in existing literature [9], formed a part of the axial codes used to categorize CHW responsibilities derived from open coding (deductive). These deductive codes specifically focused on detailing the various bridging functions of CHWs (see Section 3.2). Additional axial codes were inductively added to further categorize CHW roles that were not captured by the deductive codes but were represented during the open coding phase. Inductively added axial codes emphasized CHWs' function as direct service providers and experts in the community. Codes relating to the barriers and enablers of role performance were primarily inductively derived and thematically structured using the adapted socioecological model (see Fig 1).

The coding framework was then applied to all other transcripts, with revisions discussed through analytic memos and iteratively incorporated throughout data analysis. To maximize internal consistency, 50% of the transcripts were double-coded by a second analyst using the established codebook. A summary table of the organized themes, working descriptions of each, and illustrative quotations were presented to the larger transdisciplinary research team involved in the study. This team included mathematicians, clinical epidemiologists, health administrators, medical doctors, social scientists, and site staff. Feedback from these internal presentations was incorporated to refine interpretations and policy recommendations.

## 2.8. Ethics clearance

Ethics approval for the parent study was obtained from the University of the Philippines Manila Research Ethics Board (UPMREB – 2015-489-01). Ethics approval is renewed for all PPCS study activities across all pilot sites. Since none of the researchers involved in this secondary analysis had access to personally identifiable information obtained by the parent

study, the Johns Hopkins School of Public Health Internal Review Board (IRB) determined the present analysis exempt from IRB review.

## 3. Results

### 3.1. Demographic profile

Table 1 describes the socio-demographic characteristics of participants across the six focus group discussions. Of the participants, 15 were recruited from the rural site and 19 were from the remote site. Except for one short-tenure CHW, all focus group participants were female. Participants were between 28 and 75 years old at the time of recruitment. Due to the recruitment methods used by the parent study, participants' average years in service varied and ranged from 10 months to 31 years. Furthermore, most participants were married (82%), received secondary education or less (79%), and had no other form of employment or independent source of income apart from community health work (68%).

### 3.2. Roles and responsibilities of community health workers

Emergent themes underscored the multi-faceted nature of the roles and responsibilities undertaken by CHWs within their practice. These roles were identified through the analysis of participant accounts. Our findings reveal the intricate web of roles that CHWs effectively assume, namely: 1) community expert; 2) direct service provider, 3) service extender, 4) knowledge broker, and 5) social change agent. Table 2 provides an overview of these various roles assumed by CHWs and the corresponding responsibilities associated with each role.

CHWs have multifaceted and synergistic roles in augmenting community health outcomes. They are knowledgeable about their communities' health needs and adept at navigating local social networks. LGUs leverage this expertise for public health surveillance and ongoing health status monitoring of patients. CHWs' understanding of their communities complements their role as knowledge brokers, bridging the gap between communities and the healthcare system. They not only comprehend local contexts but also actively facilitate information and resource exchange within their networks. Additionally, CHWs act as social change agents and address various social health determinants in their practice. These tasks include advocating for mothers to participate in livelihood programs, monitoring for intimate partner violence, and helping community members access social welfare benefits. Cathy, a short-term CHW, describes their role as social change agents:

**Table 1. Socio-demographic distribution of focus group participants.**

| FGD Group | n | Mean Tenure | Mean Age | Marital Status | Highest Education | Other Income |
|---|---|---|---|---|---|---|
| 1. Short Tenure, Rural | 5 | 2.9 years | 46 | 1 Single<br>4 Married | 3 High school<br>2 College | 1 Yes<br>4 No |
| 2. Short Tenure, Remote | 6 | 2.2 years | 40 | 1 Single<br>4 Married<br>1 Domestic partner | 4 High school<br>2 College | 3 Yes<br>3 No |
| 3. Mid-Tenure, Rural | 5 | 5.6 years | 49 | 4 Married<br>1 Domestic partner | 4 High school<br>1 College | 2 Yes<br>3 No |
| 4. Mid-Tenure, Remote | 6 | 5.3 years | 47 | 6 Married | 6 High school | 3 Yes<br>3 No |
| 5. Long Tenure, Rural | 5 | 34.0 years | 61 | 4 Married<br>1 Widowed | 2 Elementary<br>2 High school<br>1 College | 5 No |
| 6. Long Tenure, Remote | 7 | 14.0 years | 50 | 6 Married<br>1 Widowed | 6 High school<br>1 College | 2 Yes<br>5 No |

*"Couples can have disagreements due to a woman's pregnancy. In several instances, these conflicts have led to domestic violence. Since some CHWs are also tasked to monitor cases of domestic violence, we try to de-escalate these tensions and support counselors to the extent we can."*

The role of CHWs as direct service providers, however, is relatively narrow. Their duties are primarily limited to blood pressure monitoring and administering first aid if other providers are unavailable. While a few CHWs recalled instances when they suggested diagnoses to patients, most exercised caution due to their fear of potentially providing inaccurate interpretations of medical conditions. This cautious approach led them to refer patients to the supervising midwives or the primary care physicians stationed at the central health unit. Gina, a long-tenure CHW, described:

*"When mothers message me on Facebook about their child's symptoms, I consult UpToDate [a digital clinical decision-support tool] to find what can be done to improve the child's condition. However, I stick to providing light advice. I'm afraid to interpret complex health information, as I could easily make mistakes and potentially harm the patient."*

Despite their limited clinical responsibilities, CHWs serve as health service extenders by aiding patients in accessing care, distributing medicines, and providing logistical support for public health initiatives. The distinction between CHW responsibilities inside and outside facilities is minimal, except for facility cleaning and maintenance. However, the presence of CHWs in barangay health centers exemplifies their role in extending services from the central health unit to more peripheral service areas. While these satellite centers often lack sufficient medical supplies for most care functions and CHWs are not trained to diagnose, their presence in these facilities ensures that patients can reliably reach individuals who can connect them to other providers within the healthcare provider network if needed.

### 3.3. Barriers and enablers to role performance

**3.3.1. Overview.** Using the socio-ecological model to examine the barriers and enablers experienced by CHWs in the field, this study reveals that role performance among CHWs is shaped as much by their individual capabilities as by their evolving social relations, organizational environments, and the broader community contexts in which they operate. Table 3

**Table 2. Overview of roles assumed and responsibilities performed by CHWs on the field.**

| Role | Role Description | Responsibilities Performed |
|---|---|---|
| *Community expert* | • Individuals who wield in-depth knowledge of community resources and networks | • Collect and manage health data<br>• Canvass households and monitor patients<br>• Identify at-risk populations |
| *Direct service provider* | • Individuals who administer hands-on health services | • Administer first contact care<br>• Conduct blood pressure checks and health assessments |
| *Service extender [a]* | • Individuals who facilitate access to existing health services and resources | • Facilitate patient referrals and transport<br>• Distribute medicines<br>• Provide logistic support for public health initiatives |
| *Knowledge broker [a]* | • Mediators who actively facilitate culturally appropriate communication between patients and the healthcare system | • Facilitate the exchange of information between providers and patients<br>• Translate health advice and promote health in a manner accessible to patients |
| *Social change agent [a]* | • Individuals who provide services that address social determinants of health | • Mediate in domestic disputes<br>• Participate in community clean-ups, disaster relief, and community-building programs<br>• Promote access to government aid (e.g., poverty reduction programs, food security assistance) |

[a]Adapted from Schaaf M, Warthin C, Freedman L, Topp SM. The community health worker as service extender, cultural broker and social change agent: a critical interpretive synthesis of roles, intent and accountability. BMJ Global Health. 2020;5(6):e002296. https://doi.org/10.1136/bmjgh-2020-002296.

**Table 3. Overview of barriers and enablers organized according to the socioecological model (SEM).**

| SEM Level | Barriers | Enablers |
|---|---|---|
| *Individual* | • Limited technological literacy<br>• Low self-efficacy in clinical decision-making | • Diskarte<br>• Skilled in emotional labor<br>• Cultural competence |
| *Interpersonal* | • Workplace tensions<br>• Patient resistance | • Embeddedness in local social networks<br>• Positive relationship with supervisors |
| *Organizational* | • Limited supplies and equipment<br>• Unclear delineation of roles and command chain fragmentation<br>• Inadequate financial support | • Access to clinical decision support<br>• Opportunities for professional development and training |
| *Community* | • Political patronage and lack of political support<br>• Inadequate network infrastructure<br>• Geographical and transportation barriers | • Access to social media and communication networks<br>• Supportive local governance |

provides a general overview of the barriers and enablers influencing CHW role performance described by participants in focus group discussions.

**3.3.2. Barriers and enablers to role performance.** *Individual Level*: Over the past five years, the healthcare facilities employing the CHWs enrolled in this study adopted several modernization initiatives. These interventions included the network-wide use of a centralized electronic health records system and the integration of telemedicine services into existing workflows. CHWs have struggled to adapt to the evolving demands of this new digital landscape, wherein they were asked to assist patients in accessing telemedicine services. While CHWs were adept at using mobile devices, many still experienced difficulties using word processors or operating computer accessories like keyboards or mice. Gaps in technological skills and a limited understanding of complex health concepts led them to rely on midwives for technical support. The experiences of Lydia, a long-tenure CHW, highlighted such issues:

*"We're used to recording data using pen and paper. In my case, I don't know how to use MS Excel or how to type on a keyboard properly. It takes me an hour to type anything."*

CHWs demonstrated several individual-level capacities that bolstered their role performance. These enablers include *diskarte*, their skill in emotional labor, and their cultural competence. When confronted with resource deficits, CHWs employed self-devised strategies or their *diskarte* to navigate emergent challenges. *Diskarte* can be defined as an individual's ability to overcome situational limitations to ensure survival. However, unlike resilience or resourcefulness, its use denotes not only a lack of resources but also unequal social positions [26]. This usually involves formulating strategies to improve a situation despite the limitedness of support. In healthcare settings, its the poor implementation of policies that adequately secure resources needed by the workforce to address the concerns of patients contributes to its repeated use [27]. Drawing from the data, one CHW described how they operationalized *diskarte*:

*"Barangay officials are always angry when we request materials. They often say that it's our own diskarte [strategy] on how to handle this problem. If we can't manage to sort it out, we take money from the donation box. We use our allowances and we pool funds with the nurses and midwives. We rarely get reimbursed for this."*

CHWs' skill in performing emotional labor—defined as the deliberate management of one's emotions to meet occupational expectations [28,29]—was identified as an individual-level enabler. When confronted with stressful encounters with their colleagues or patients, the ability to suppress outward emotions was perceived by CHWs as essential for maintaining professional relationships and ensuring effective service delivery:

*"We need to maintain a calm and personable front. We have to stop ourselves from outwardly expressing frustration. If not, patients will not listen to us or cooperate." (Tina, short-tenure CHW)*

Lastly, cultural competence or one's ability to understand "the importance of social and cultural influences on patients' health beliefs and behaviors" [30], emerged as a role enabler. The relevance of cultural competence came to the fore whenever CHWs interpreted patients' beliefs on alternative medicine and discerned unspoken cues in communication. Drawing on their own lived experiences, including navigating past medical controversies like the Dengvaxia vaccine rollout—which involved concerns over its safety and efficacy—CHWs could understand and position community attitudes towards perceived alternative treatments (e.g., the use of Ivermectin and steam inhalation against COVID-19) within these broader historical contexts. Furthermore, their adeptness in perceiving nuanced shifts in body language, facial expressions, or the linguistic subtleties in patients' choice of words and tone was instrumental in identifying instances where patients withheld critical information. As one CHW stated:

*"You'll need to be attuned to patients' emotions because sometimes they don't share their concerns with their providers outright. You'll only grasp their needs by picking up on these cues through your interactions with them." (Jane, short-tenure CHW)*

**Interpersonal Level:** Navigating social relationships within community health work presented challenges to role performance, mainly through tensions with fellow CHWs or the patients they serve. Interpersonal workplace tensions surfaced when CHWs encroached upon the catchment areas of their peers, a scenario described by Cathy:

*"Some CHWs over-involve themselves in tasks outside their assignment areas. They bypass you and interfere with your work. If they have a problem with how you manage your assigned area, they should consult with you first."*

Nearly all participants reported some form of resistance from patients. Tense exchanges often arose within the context of vaccine hesitancy or whenever CHWs needed to persuade patients to adhere to treatment plans. For CHWs assigned to more affluent catchment areas, patient resistance was potentially influenced by underlying class tensions. Since most CHWs came from lower-income backgrounds, including participants in this study, these subtle tensions were felt in the ways their access to gated spaces was restricted and in the pressure they felt to exhibit more deference towards wealthier patients. As expressed by Cathy:

*"My assigned area is one of the wealthier areas in the municipality. When I introduced myself as a CHW from the municipality, they refused to let me in their homes. They refuse even to have their blood pressure checked and they'd always ask to see my ID."*

Two enablers of role performance were noted at an interpersonal level, namely: embeddedness in community networks and social support from supervisors and fellow CHWs. CHWs' embeddedness in local community networks was characterized not merely by their in-depth knowledge of community members' lives but also by their ability to navigate socio-cultural dynamics, familial structures, and dominant health narratives within the community. Such embeddedness enabled CHWs to identify or predict health-related needs. Community embeddedness was reflected in the use of the term *Marites* – a caricature denoting women who are over-involved in others' affairs – employed by CHWs as a self-descriptor. While the term 'Marites' carried negative connotations associated with the spread of gossip, the term was also employed positively as it signified one's extensive knowledge of the community. As Linda, a mid-tenure participant, describes:

*"They call us 'Marites'. We know everything about the community. We know people's marital status, sexual orientation, and family histories. Because of this, we can identify who hasn't been vaccinated, who needs medication, or who should attend family planning classes."*

Lastly, under the guidance of supervisors and bolstered by the camaraderie among fellow CHWs, participants felt more prepared to perform their roles. This was evident when midwives offered their technical expertise to help CHWs provide patient advice and when CHWs highlighted the importance of team cohesion for effectively gathering data across neighborhoods. In the words of Grace, a short-tenure participant:

*"Building team solidarity with your fellow CHWs is crucial. Without a strong rapport, all these reports won't come together. It will negatively impact the entire community. Supporting each other is important since not all CHWs pursued higher education."*

***Organizational Level***: At an organizational level, the scarcity of medical supplies and financial limitations were the most frequently reported barriers to role performance. Participants cited that blood pressure monitors, medicines, technological devices, and even office supplies were often in short supply, as Gina (short-tenure CHW) notes:

*"We have limited supplies so we circulate nebulizers. We need to disinfect them but when we request them back, we discover our patients also lend the nebulizers to their neighbors. The nebulizers go missing since so many people borrow them."*

In addition to material constraints, a fragmented command structure and ambiguous scope of their roles impeded CHWs' capacity to perform their duties optimally. CHWs reported being accountable to various authorities, including midwives, nurses, the CHW federation, and barangay captains. Due to the lack of coordination among the overseers mentioned, tasks often overlapped and became redundant:

*"The midwife typically assigns our tasks, but we also take assignments from the nurses. Occasionally, we also receive tasks from the CHW federation. When we receive orders from different sources, we try to assess if they overlap or else it duplicates." (Gina)*

At times, unclear command structures result in CHWs assuming responsibilities typically designated to other public servants. The unclear lines of authority made it challenging for CHWs to discern to whom they were accountable. Furthermore, the multitude of actors delegating tasks led to the overextension of their roles, as CHWs found themselves shouldering duties typically assigned to individuals higher up in the command chain. As Marie, a mid-tenure CHW, states:

*"I hope they can reduce our workload. The responsibilities of barangay council members should be done by them. We voiced our concerns. Before, even the distribution of national IDs was delegated to us… even teachers' duties were assigned to us."*

Amidst the organizational barriers experienced, training opportunities and the availability of clinical decision software (e.g., UpToDate) emerged as crucial support mechanisms for CHWs. Sessions focusing on using technological tools, conducting patient screenings, and engaging in scenario-based collaborative care exercises were essential in enhancing health literacy and equipping CHWs with skills for effective practice. Tanya, a long-tenure CHW, expressed that these sessions helped her understanding of various illness conditions:

*"Training seminars enhance our understanding of various disease conditions. For example, training allowed us to distinguish between the presentation of chickenpox and measles symptoms."*

Experiences of CHWs who frequently used UpToDate also reported that clinical support tools augmented self-efficacy in the field. The decision to perform medical interventions remains the responsibility of physicians. However, tools such as UTD enhance their knowledge-base on various diseases while learning about non-pharmacological measures to help the patient. Where other CHWs may have sought directions from a midwife, frequent UpToDate users described that they were able to rely on their own decision-making with the help of these tools:

*"UpToDate is useful, particularly during staff shortages. It empowers us to provide care even in the absence of midwives. These resources improve our ability to address patient needs." (Karen, mid-tenure CHW)*

***Community/ Policy Level***: Leadership that prioritized health initiatives reportedly enhanced participants' access to necessary supplies. Conversely, political patronage and corruption endangered job security and compromised the integrity of data reporting. The distribution of government aid relied on the records of low-income households submitted by CHWs within a community. Political patronage has been observed, stemming from first-hand accounts of local officials influencing decisions on which households are to be included or excluded from these lists. Leni, a mid-tenure CHW, describes one of such accounts:

*"I hope they don't purposefully alter the documents we provide. There was a time when a publicly-sponsored community activity only lasted for a week. [The local official] ordered me to make it appear as if it lasted for a month on paper. I said this was very difficult to do and questioned if this can even be documented since it never happened as they described."*

Apart from local governance, geographical barriers and weak network infrastructure hindered service delivery. Constant flooding and mountainous terrain often made routine immunizations challenging due to staff shortages in most communities. While CHWs' use of social media (i.e., Facebook Messenger) has bridged some of these gaps, cellular and internet services are still unreliable for many. With the recent introduction of electronic health records and telemedicine platforms in both municipalities, the effects of infrastructural deficiencies were even more pronounced as Dane, a short-tenure CHW, described:

*"There's no cellular signal that reaches our health center. It goes without saying, we don't have WiFi. When I'm on duty, patients are unable to reach me. It's frustrating because I can't use UpToDate or make telemedicine referrals."*

## 4. Discussion

### 4.1. Community health workers at the fulcrum of health communication

The objectives of this study were twofold: 1) to explore the roles and responsibilities of CHWs; and 2) to identify the perceived barriers and enablers they encounter in performing their roles. Insights gathered from focus group discussions in remote and rural areas of the Philippines highlighted the extensive responsibilities shouldered by CHWs. CHWs assumed clinical and non-clinical roles within their practice – ranging from direct service provision, navigating patients through healthcare processes, to intervening on the social factors that influence health outcomes. Barriers to role performance included limited technological literacy, low self-efficacy in clinical decision-making, workplace tensions, patient resistance, inadequate supplies, unclear role delineation, and political instability. Conversely, enablers encompassed strategies such

as *diskarte*, proficiency in emotional labor, cultural competence, strong interpersonal relationships, organizational support through clinical decision aids and training opportunities, and community support via social media and engaged local governance.

Echoing previous findings, this study underscored the indispensable function of CHWs in facilitating the bidirectional flow of information between patients and the healthcare system [1,9]. Their mediating role is especially evident when relational hierarchies exist between patients and providers or between community members and local government officials. CHWs bridge patient-provider communication given their ability to translate complex health information in a language and register accessible to patients. Conversely, CHWs were able to advocate for the needs of community members given their proximity to both healthcare providers and government officials alike.

CHWs' embeddedness in community and healthcare provider networks supports their bridging functions and positions them at the fulcrum of health communication. The physical proximity and relational embeddedness of CHWs to the community intimated an in-depth understanding of local beliefs, practices, and behavior patterns. Conversely, the liminality of the CHW position—straddling the line between community member and healthcare professional—afforded CHWs access to information otherwise inaccessible to community members. Through effective training and support, CHWs have the potential to optimize human healthcare resource allocation and maximize local capacities in underserved settings.

The explicit boundaries of CHW responsibilities were ambiguous and described by participants as overextending the scope of health service delivery. For example, disseminating social service benefits and mobilizing disaster evacuation efforts also depended on CHW involvement. Fragmentation in the command chain and the absence of a formal job description contributed to such role ambiguity [31]. Overlapping oversight from multiple supervisory bodies and the lack of standardized workplace protocols meant that CHWs often handled an assortment of seemingly unrelated tasks. Within the data, this was corroborated by accounts where participants felt unable to refuse ad hoc tasks or felt overwhelmed by the concurrent tasks assigned to them by the midwife, the CHW federation, and local officials.

## 4.2. The precarity of community health work

CHWs were implicitly tasked with navigating the complexities of shifting medical environments, divergent political and societal expectations, and resource constraints faced when performing their roles. Material exigencies and the lack of organizational and policy-level support heightened the precarity of their positions. Although expected to assume multiple roles within the healthcare system, a significant fraction of their tasks were neither formally accounted for nor compensated. CHW remuneration hinged on locally appropriated health budgets [6] and the efficiency of local administrators in managing bureaucratic procedures [19]. When health budgets were sidelined or onboarding was delayed, CHWs expectedly experienced compensation issues. This resulted in CHWs receiving no payment, reduced payment, or enduring delays that could span a few months to over five years. In one of the more severe cases noted in this study, changes in local leadership reportedly led to the abrupt dismissal of an entire cadre of CHWs due to their perceived affiliations with the previous administration. However extreme this may seem, parallel accounts have been documented in past research examining the same phenomenon of interest [1,4].

The lack of workplace protections typically afforded to other healthcare roles heightened the precarity of CHW work, as previous literature suggests [6,32]. Beyond timely remuneration, study participants were neither formally entitled to standardized employment benefits nor did they receive severance pay or retirement benefits. Previous research has consistently shown the profound impact of non-financial incentives on CHWs' professional motivation. Access to training and a driving sense of purpose for the job propelled CHWs to remain in service even in the absence of remuneration [32,33]. However, the unpredictability of remuneration made CHW work untenable as a primary income source. In most instances, participants had to depend on their partners or other family members to support their daily needs.

The absence of firm entry requirements within the profession has disproportionately driven economically insecure women to engage in community health work [32]. This trend can largely be attributed to the scarcity of viable employment

options available for individuals with limited education and the child-rearing obligations borne by many of the women interviewed. In environments rife with poverty, the potential for the exploitation of these women is of grave concern. Societal norms reinforce the expectation for women to assume caring roles both domestically and professionally. Furthermore, the appeal of CHW programs to individuals who might otherwise struggle to find employment has, historically, been leveraged to justify unpaid labor [32–34]. The pervasive issues that underlie community health work were best exemplified in the Philippines' Department of Health pocket handbook for CHWs, which demanded from CHWs unwavering "perseverance [when performing their roles] despite difficulties, setbacks and lack of financial support" [35]. This has situated CHWs in a precarious position where their labor is institutionally undervalued, yet a deep commitment and altruistic 'love' for their work is expected in return.

### 4.3. Overcoming situational limitations through diskarte and social capital

A systematic review of the accessibility of essential medicines in low-middle-income countries indicated that, on average, CHWs were confronted with difficulties in fulfilling supply-side demands about one-third of the time [36]. These findings aligned with local literature on CHW experience, wherein workforce shortages and the unavailability of medical supplies have impeded health service delivery [6,33]. In the present study, managing the burden of resource scarcity was often shifted onto individual and interpersonal capacities in the field. Demands unmet on an organizational or community level were predominantly addressed by CHWs through their own *diskarte* or self-devised strategy.

Social capital emerged as a critical enabler for CHW role performance amidst situational limitations. Following Bourdieu, social capital can be defined as the sum of "actual or potential resources which are linked to [the] possession of a durable network of more or less institutionalized relationships of mutual acquaintance and recognition" [37]. Social capital underpinned CHWs' ability to effectively mobilize resources, assist in public health surveillance efforts, and mediate between patients and the health system. Previous research similarly suggested that the effectiveness of CHWs in the field is largely affected by the social capital they cultivate with community members. A cross-sectional study conducted during the COVID-19 pandemic reported that CHWs with stronger community ties were more likely to participate in pandemic response activities [38].

Within the present study, the role of social capital was even more apparent when considering its interplay with *diskarte*. Where *diskarte* reflects an individual's ability to navigate and overcome resource constraints, network embeddedness cultivated through social capital enhances this ability. Examples such as pooling donations from fellow CHWs to address the lack of medical supplies or deciding to create Facebook group chats to connect with patients in remote areas were some ways social capital factored into participants' *diskarte*. Their profound understanding of local resources and networks can potentially improve the effectiveness of program interventions if these strategies were accounted for. Although the conceptual relationship between *diskarte* and social capital remains underexplored, this study underscored the potential of exploring this milieu in future research and policy. Productively scaling out *diskarte* could involve broadening CHW participation in health decision-making and program design. Specifically, this could involve initiating regular sensemaking workshops with CHWs to obtain feedback to appraise and evaluate existing public health policy on both local and national levels.

### 5. Policy recommendations

This study underscored the pressing need to clearly define the role of CHWs within the healthcare provider network, address challenges faced in the field, and amplify existing workforce capacities. While the Magna Carta can undoubtedly provide the legal basis for workplace protections [13], it is imperative to develop mechanisms that allow CHWs to assert these protections actively. In practice, CHWs could be oriented on these legal protections and the implications of the Magna Carta in their everyday practice. Training could not only cover their rights, but also offer practical examples of potential rights violations and guidance on where they might be able to safely report these concerns. Additionally,

establishing a credentialing system could formalize and affirm the professional status of CHWs, thus reinforcing their rights and roles in the healthcare system. The lack of financial stability among CHWs was a core finding of this study. An organized credentialing system can pave the way for CHWs to receive standard compensation that is commensurate with their efforts. Regular collaboration meetings between BHWs and local health committees are also encouraged. This can lay out the ideal budgetary allotment for future projects and prevent CHWs from spending out-of-pocket. Such meetings can also help establish a formal reimbursement system for unforeseen expenditures. These measures are vital for empowering CHWs to fully benefit from the protections granted by the Magna Carta and to overcome implementation gaps in past policies.

A well-supported implementation of the Universal Healthcare Law may open a lot of doors for CHWs. With health promotion at its core [14], funding for strengthened primary care services can facilitate increased training opportunities. Knowledge and skill strengthening opportunities include introducing clinical decision support systems like UpToDate, hands-on training for digital literacy skills, and encouraging interprofessional collaboration. Beyond the traditional lecture set-up, regular group learning opportunities are encouraged. CHWs work in teams. Group activities help build their dynamic while allowing them to process novel information with individuals who have shared experiences with patients. Reinforcing knowledge with practical skills is also a good step forward as it allows CHWs to see how lectures can apply in their unique settings. Continuous provision of multimodal opportunities is advantageous as it allows workers to pattern their everyday practice to available evidence. These efforts establish their professional identity while allowing them to reach more people in their respective communities. As previous studies have suggested, supporting CHWs in critical decision-making can further integrate their roles within the local health system [1,39]. This study builds upon prior findings by specifying strategies to improve CHW participation within the healthcare provider network.

## 6. Scope and limitations

Conducted in rural and remote municipalities in the Philippines, the reported results offer insights into CHW experiences in resource-limited settings. While these findings are valuable for identifying gaps in existing CHW policy, the contextual specificity of the study must be acknowledged. The sample was drawn from a population that had undergone various health system-strengthening interventions over six years preceding the focus group discussions. Interventions included introducing unified electronic health record systems, universal health coverage, UpToDate training, and developing telemedicine capabilities. The narratives in this study may not fully represent the breadth of challenges CHWs encounter in areas where such interventions are still absent or in their nascent stages. Although these results may not be immediately generalizable to all CHW experiences, the gradual implementation of the Philippines' Universal Healthcare Law and its associated system-strengthening initiatives may increase the transferability of our findings as these interventions become more widespread over time.

Employing a secondary analysis on pre-existing data inherently confines the study's scope to the parameters established by the parent program. Although the facilitator's guide from the parent study covered the objectives of the current research, topics excluded in the primary data collection remain beyond the scope of this analysis. Subsequent work can build on this research by elaborating on the recruitment strategy of the parent study. This study intentionally recruited participants by tenure and place of assignment. Yet, in exploring their narratives, no distinct variation in experiences emerged across tenure lengths or between rural and remote locales. The uniformity in these accounts might suggest that the barriers and enablers influencing role performance are deeply entrenched, transcending both time and geographical boundaries. However, it is also conceivable that the qualitative depth of this study, potentially constrained by time and the interview guide used, may not have sufficiently captured the subtle differences between these categories. Future studies could incorporate questions that probe into the impact of tenure and place-based factors on CHW experience. Examples of potential questions may comprise inquiries into how participants' roles have evolved over time and across various administrations, their perceptions of fellow CHWs who are either newer or more experienced, and the barriers and enablers they perceive to be unique to their community context.

## 7. Conclusion

This research explored the roles of CHWs, the diverse challenges they face in their field of practice, and the enabling factors that allowed them to overcome situational limitations effectively. Adopting Bronfenbrenner's socioecological model as a framework, focus group data revealed that factors at multiple levels influence CHW performance. Their multi-faceted bridging roles were critical yet precariously positioned. Faced with resource scarcity and ever-evolving political contexts, CHWs demonstrated remarkable resilience through their *diskarte* or self-devised strategy and by maximizing support accessed from their social networks. These individual and interpersonal capacities were augmented with the availability of clinical decision support systems, network connectivity, and training opportunities.

The findings of this study echo prior research that underscored the need for added workplace protections for CHWs. While high-level efforts to expand such policies are at the cornerstone of change, a multi-level approach is still required to overcome inertia at lower levels of the system. Programmatic interventions should also focus on mending command chain fragmentation at the operational level, enacting mechanisms to manage power asymmetries, and fostering digital literacy through hands-on training. Bridging the gap between theory and practice, and translating policies into tangible action, are enduring challenges for health systems to address. Recognizing the vital roles assumed by CHWs and the myriad of challenges they navigate, the lived experiences outlined in this study forward a perennial appeal for more responsive CHW programs worldwide.

## Supporting information

**S1 Appendix. Semi-Structured Interview Guide.**
(DOCX)

**S1 File. Remote Site Transcripts.**
(ZIP)

**S2 File. Rural Site Transcripts.**
(ZIP)

## Acknowledgments

The authors sincerely thank the CHWs who shared their invaluable experiences, providing the foundational narratives for both the parent study and the secondary analysis detailed in this paper. We also wish to acknowledge the Philippine Primary Care Studies program (also known as the Program on Health Systems Development under the University of the Philippines Center for Integrative and Development Studies) for granting access to the focus group data collected between June and July 2023. Lastly, we express our utmost gratitude to the data collectors of the parent study for recruiting participants and facilitating each focus group session.

## Author contributions

**Conceptualization:** Regine Ynez H. De Mesa.

**Formal analysis:** Regine Ynez H. De Mesa, Anton Elepaño.

**Investigation:** Noleen Marie C. Fabian, Johanna Faye E. Lopez.

**Methodology:** Regine Ynez H. De Mesa.

**Supervision:** Leonila F. Dans, Antonio Miguel L. Dans.

**Writing – original draft:** Regine Ynez H. De Mesa, Noleen Marie C. Fabian.

**Writing – review & editing:** Regine Ynez H. De Mesa, Zoé Mistrale Hendrickson, Carol Stephanie C. Tan-Lim, Carl A. Latkin, Mia P. Rey.

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
