## [Decision Letter · Decision Letter 0]

17 Oct 2024

PGPH-D-24-01354

An Underpaid Labor of Love: A Multi-Level Analysis of the Barriers and Enablers to Role Performance Among Community Health Workers in Rural and Remote Municipalities in the Philippines

Dear Dr. De Mesa,

Thank you for submitting your manuscript to PLOS Global Public Health. After careful consideration, we feel that it has merit but does not fully meet PLOS Global Public Health’s publication criteria as it currently stands. Therefore, we invite you to submit a revised version of the manuscript that addresses the points raised during the review process.

The submission has been reviewed by 3 reviewers, and their comments are provided below. Concerns mostly relate to the reporting of the setting and contextualization of the study. You mention two appendices, including a copy of the interview guide. Please ensure these are submitted with your revised manuscript.

We look forward to receiving your revised manuscript.

Kind regards,

Marianne Clemence

Staff Editor

Journal Requirements:

 1. Please include a complete copy of PLOS’ questionnaire on inclusivity in global research in your revised manuscript. Our policy for research in this area aims to improve transparency in the reporting of research performed outside of researchers’ own country or community. The policy applies to researchers who have travelled to a different country to conduct research, research with Indigenous populations or their lands, and research on cultural artefacts. The questionnaire can also be requested at the journal’s discretion for any other submissions, even if these conditions are not met.  Please find more information on the policy and a link to download a blank copy of the questionnaire here: https://journals.plos.org/globalpublichealth/s/best-practices-in-research-reporting. Please upload a completed version of your questionnaire as Supporting Information when you resubmit your manuscript. 2. Please amend your detailed Financial Disclosure statement. This is published with the article. It must therefore be completed in full sentences and contain the exact wording you wish to be published. **Please only choose the relevant sentences from below** 1. Please clarify all sources of funding (financial or material support) for your study. List the grants (with grant number) or organizations (with url) that supported your study, including funding received from your institution. 2. State the initials, alongside each funding source, of each author to receive each grant.3. State what role the funders took in the study. If the funders had no role in your study, please state: “The funders had no role in study design, data collection and analysis, decision to publish, or preparation of the manuscript.”4. If any authors received a salary from any of your funders, please state which authors and which funders. If you did not receive any funding for this study, please simply state: “The authors received no specific funding for this work.” 3. In the online submission form, you indicated that "Data can be made available by the authors upon reasonable request.".  All PLOS journals now require all data underlying the findings described in their manuscript to be freely available to other researchers, either 1. In a public repository, 2. Within the manuscript itself, or 3. Uploaded as supplementary information. This policy applies to all data except where public deposition would breach compliance with the protocol approved by your research ethics board. If your data cannot be made publicly available for ethical or legal reasons (e.g., public availability would compromise patient privacy), please explain your reasons by return email and your exemption request will be escalated to the editor for approval. Your exemption request will be handled independently and will not hold up the peer review process, but will need to be resolved should your manuscript be accepted for publication. One of the Editorial team will then be in touch if there are any issues. 4. Please provide separate figure files in .tif or .eps format. For more information about figure files please see our guidelines: https://journals.plos.org/globalpublichealth/s/figures https://journals.plos.org/globalpublichealth/s/figures#loc-file-requirements 

Additional Editor Comments (if provided):

Reviewers' comments:

Reviewer's Responses to Questions

**Comments to the Author**

1. Does this manuscript meet PLOS Global Public Health’s publication criteria ? Is the manuscript technically sound, and do the data support the conclusions? The manuscript must describe methodologically and ethically rigorous research with conclusions that are appropriately drawn based on the data presented.

Reviewer #1: Yes

Reviewer #2: Partly

Reviewer #3: Yes

2. Has the statistical analysis been performed appropriately and rigorously?

Reviewer #1: Yes

Reviewer #2: N/A

Reviewer #3: Yes

3. Have the authors made all data underlying the findings in their manuscript fully available (please refer to the Data Availability Statement at the start of the manuscript PDF file)?

Reviewer #1: No

Reviewer #2: Yes

Reviewer #3: Yes

4. Is the manuscript presented in an intelligible fashion and written in standard English?

Reviewer #1: Yes

Reviewer #2: Yes

Reviewer #3: Yes

5. Review Comments to the Author

Reviewer #1: This is is a well written manuscript providing evidence on known successes and challenges regarding the performance of CHWs.

I do not have major substantive comments for the others, just a couple of suggestions for consideration to improve the manuscript.

The first is to fine more updated numbers for the CHWs rather than the 2015 that is referred to in the background if available.

The second is to put some of the context content that is in the policy recommendation earlier in the documents so that the section focuses more on what the authors are recommending. currently, these are buried in the explanations

Reviewer #2: This is an interesting paper on an important topic that is currently receiving a lot of international attention again. Key comments for revisions:

1. The authors explain that this study is based on the secondary analysis of data collected for a parent program piloted a study on a clinical decision support tool for CHWs. However, it does not become clear in the methods how the original data were read and analysed to answer a different set of questions and objectives for this paper. Both the abstract and the methods section of the main paper need to elaborate on this.

2. As a reader from another region of the world I would need a brief overview of the structure and levels of the Philippine health system to be able to understand how CHWs are located within it.

3. There appears to be contradictory information about the scope and role of CHWs: the introduction mentions literature that says that they can only work under direct supervision from nurses and midwives, yet the main text paints a much more diverse and possibly fragmented picture. This would need clarification, including an explanation of how the UpToDate tool is changes CHWs' scopes and roles.

4. It is unclear how the role descriptions in table 3 were generated, i.e. whether they are self-reported by CHWs. Overall the evidence base for the findings presented remains quite opaque to the reader.

5.The discussion currently continues to present result. It does not constitute a discussion of presented findings.

6. A minor issue: "Diskarte" is a term that is unfamiliar to me and would need to be explained.

Reviewer #3: This is a very nice study reporting the insights and opinions of community health workers in the Philippines. I actually find the paper incredibly well done and very interesting; I was fascinated with the analysis, and learned a number of new ideas.

In particular, I appreciate the authors introducing a new audience to the concept of “diskarte.” This term points towards something I have seen in community health programs across multiple continents, even if the word itself is from a local language.

Similarly, I appreciated learning about the term “marites.” I was fascinated that it held both positive and negative connotations, and that the community health worker who shared this term wore it as a (perhaps embarrassing?) badge of honor. This duality mirrors the contradiction that many community health workers are forced to live - that their jobs bring both positivity and negativity into their lives. But this ability to be positioned as a “marite” is likely a product of their personalities and talents in being able to relate with people interpersonally. It likely also comes from an inborn curiosity, and willingness to serve. As has previously been said, “community health workers are not trained, they are born.” Only the “marite” can get that deep into the community, and this position brings back new functionality to the health system.

Otherwise, the paper reported on dynamics that have been well described in countless other papers, though the authors do a good job in explaining how all this works in the Filipino context.

The only suggestions that I would recommend, with the aim of further improving the paper, are as such:

• I recommend that the authors consider a new title. When I first read the title, I found myself not interested in reading further. I doubted whether or not we needed another paper looking at barriers and enablers in community health. The term “an unpaid labor of love“ also does not fully capture what they are trying to report, because it sounds like because it is “a labor of love”, it can (should?) remain underpaid. Maybe they can find a better title by leaning into one of the insights I mention aboveBasically, what I think this paper adds is a very good overview of the Filipino context, and uniquely local conceptualizations of how the Filipino woman is situated in health work (marite) and how they bring to their jobs incredible ingenuity (diskarte).

• Similarly, the abstract does not capture the analytic depth or complexity of the study. For example, the authors claim that this study is necessary because “barriers and enablers… remain underrepresented in existing literature.“ This is simply not true, as this topic has been extensively studied. What this study actually is doing is pulling from the Filipino context unique insights that are relevant for community health workers everywhere, and their multinational struggle for professionalization. I recommend the authors look at multiple other articles reporting on "barriers and enablers," and to find language not common in those papers. For example, when the authors mention in the abstract about the need for "legal safeguards," it sounds cliched; what they are describing is the incredible experience of decades of existing laws being poorly implemented. This inability to make existing community health laws have teeth invaluable information for the current struggle to support community health everywhere, but a reader wouldn't know that just looking at the abstract. I also would avoid the term "unsustainable;" what I hear is that the position of a community health worker in the Philippines HAS been sustained for decades, but it is just poorly supported and poorly positioned. I could go on line-by-line, but this is true for most of the abstract; basically, the paper is much more complex, nuanced and original, and I wish the abstract mirrored that better.

• Finally, while I think that the authors are very well read in community health, there is a wide variety of activity that they should know about.

o I refer them to the National C3 Council effort, to further define CHW roles, which mirrors and somewhat built upon what they report. https://www.c3council.org

o I also refer them to the work of the Community Health Impact Coalition (CHIC), which is a multinational coalition of non-governmental organizations working to make professionalized community health workers the norm worldwide. Many of the publications produced by this group at CHIC may be helpful for supporting this current analysis, or future analyses. www.joinchic.org

6. PLOS authors have the option to publish the peer review history of their article (what does this mean? ). If published, this will include your full peer review and any attached files.

**Do you want your identity to be public for this peer review?** For information about this choice, including consent withdrawal, please see our Privacy Policy .

Reviewer #1: No

Reviewer #2: No

Reviewer #3: **Yes: ** Daniel Palazuelos, MD, MPH

---

## [Decision Letter · Decision Letter 1]

22 Jun 2025

PGPH-D-24-01354R1

Resilience, Ingenuity, and Identity: A Multi-Level Analysis of the Filipino Community Health Worker Experience in Rural and Remote Municipalities in the Philippines

Dear Dr. De Mesa,

Thank you for submitting your manuscript to PLOS Global Public Health. After careful consideration, we feel that it has merit but does not fully meet PLOS Global Public Health’s publication criteria as it currently stands. Therefore, we invite you to submit a revised version of the manuscript that addresses the points raised during the review process.

The manuscript has been evaluated by two reviewers, and their comments are available below. The reviewers are largely satisfied with your paper but reviewer 4 has raised some minor issues regarding clearly stating the research gap the manuscript addresses. 

Could you please revise the manuscript to carefully address these comments?

We look forward to receiving your revised manuscript.

Kind regards,

Jenna Scaramanga

Staff Editor

Journal Requirements:

Additional Editor Comments (if provided):

Reviewers' comments:

Reviewer's Responses to Questions

**Comments to the Author**

1. If the authors have adequately addressed your comments raised in a previous round of review and you feel that this manuscript is now acceptable for publication, you may indicate that here to bypass the “Comments to the Author” section, enter your conflict of interest statement in the “Confidential to Editor” section, and submit your "Accept" recommendation.

Reviewer #3: All comments have been addressed

Reviewer #4: (No Response)

2. Does this manuscript meet PLOS Global Public Health’s publication criteria ? Is the manuscript technically sound, and do the data support the conclusions? The manuscript must describe methodologically and ethically rigorous research with conclusions that are appropriately drawn based on the data presented.

Reviewer #3: Yes

Reviewer #4: Yes

3. Has the statistical analysis been performed appropriately and rigorously?

Reviewer #3: I don't know

Reviewer #4: Yes

4. Have the authors made all data underlying the findings in their manuscript fully available (please refer to the Data Availability Statement at the start of the manuscript PDF file)?

Reviewer #3: Yes

Reviewer #4: Yes

5. Is the manuscript presented in an intelligible fashion and written in standard English?

Reviewer #3: Yes

Reviewer #4: Yes

6. Review Comments to the Author

Reviewer #3: Great job responding to the comments. I look forward to being able to share this paper with colleagues.

Reviewer #4: The manuscript is well organized, specially after peer review process. But still it needs to be re stated the research gap, it is weak to my perspective. Also consider to explain why you use this method to address this gap.

7. PLOS authors have the option to publish the peer review history of their article (what does this mean? ). If published, this will include your full peer review and any attached files.

**Do you want your identity to be public for this peer review?** For information about this choice, including consent withdrawal, please see our Privacy Policy .

Reviewer #3: **Yes: ** Daniel Palazuelos

Reviewer #4: No

---

## [Editor Report · Decision Letter 2]

8 Jul 2025

Resilience, Ingenuity, and Identity: A Multi-Level Analysis of the Filipino Community Health Worker Experience in Rural and Remote Municipalities in the Philippines

PGPH-D-24-01354R2

Dear Ms De Mesa,

We are pleased to inform you that your manuscript 'Resilience, Ingenuity, and Identity: A Multi-Level Analysis of the Filipino Community Health Worker Experience in Rural and Remote Municipalities in the Philippines' has been provisionally accepted for publication in PLOS Global Public Health.

Best regards,

Julia Robinson

Executive Editor